# The Electrification of Ships Using the Northern Sea Route: An Approach

**Christophe Savard [1,*,†] , Anni Nikulina [2,†], Céline Mécemmène [1] and Elizaveta Mokhova [2]**

[1]  Mainate Labs, 16 rue Notre Dame de l'Oratoire, 43270 Allègre, France; cme@mainate.com
[2]  Department of Organization and Management, Saint-Petersburg Mining University,
   199106 St-Petersburg, Russia; nikulina_ayu@pers.spmi.ru (A.N.); s180819@stud.spmi.ru (E.M.)
[*]  Correspondence: cjs@mainate.com
[†]  These authors contributed equally to this work.

**Abstract:** Global warming is causing a major ice retreat from the North Pole. From now on, this retreat allows almost permanent movement between East and West off the coast of the Russian Federation along the Northern Sea Route (NSR). For a long time, navigators have been trying to use this route which significantly reduced the distance between continents. The amount of freight that currently travels on the NSR will inevitably increase in the coming years. To reduce environmental risks, one possible option is not to supply ships with heavy fuel oil. The ships could then be electrically powered and navigate in stages from one port to another along the route to refuel for energy. This electrical energy can be produced on site from renewable energy sources. In this article, a first feasibility analysis is outlined, taking into account the tonnage constraints for navigating on a possible route for the NSR, the cost of energy production and the possible location of several ports of call. Under current economic conditions, the solution would not be profitable as it stands, but should become so at a later stage, which justifies starting to think about a future full electrification of navigation on the NSR, which will also contribute to the economic development of the Russian Federation northernmost regions.

**Keywords:** electrical energy; Northern Sea Route; Arctic maritime traffic

---

The year round usage of the Northern Sea Route (NSR) to transport intercontinental freight will soon be a reality. In order to preserve the global environment, it seems appropriate to consider a development that reduces the environmental impact of maritime traffic in the Arctic at a time when it is beginning to develop considerably.

## 1. Connect Europe to Asia through the Arctic Ocean

It is only now that this idea, which is ancient, is really becoming a reality, as a collateral benefit of global warming. It has been a long-standing question of using the Arctic Sea to connect East and West.

### 1.1. Conquest of the Sea Route by the North of the Russian Coast

Currently, infrastructure is not yet sufficiently developed along the Northern Sea Route to allow for the sustainable establishment of a trade route between Asia and Europe via the northern part of the Russian Federation [1]. However, there is an opportunity to make proposals in line with history, namely to support the development of this new seaway while reducing the environmental impact of its use.

The Arctic is a territory rich in natural resources, with strategic geopolitical importance. The Arctic's energy resource potential is a vector for the region's development, associated with

land and maritime transport and logistics infrastructure [2]. According to the Minister of Natural Resources of the Russian Federation Dmitry Kobylkin, oil reserves of the Russian Arctic zone are 7.3 billion tons, natural gas, i.e., about 55 trillion cubic meters [3]. At the same time, more than 60% of the recoverable hydrocarbon resources of Russia are concentrated in the Arctic (260 billion tons of standard fuel) [4]. Around this new seaway, in addition to the potential for international navigation between East and West, there are also potential conflicts over the still unexploited resources of both oil and natural gas.

In the Arctic, the impact of global warming is twice as rapid as elsewhere. This is particularly the case in Spitsbergen, but not only. Russia's northern coasts are now experiencing ice-free summers. As a result of global warming, the retreat of ice in the Arctic allows the opening of a summer waterway along more than 5600 km of Russian Federation [5] and 2400 km of Norwegian coastline for increasingly long periods each year.

In 2050, forecasters expect that the northern shipping route, the NSR along the Russian coast, will be completely ice-free [6]. Two other routes through the Arctic Ocean are possible to connect Europe and Asia by sea: the North West Route (NWR), which runs along the Canadian coasts, and the direct passage through the pole, which is the shortest route: the North Pole Route (NPR). Indeed, in the long term, if the ice continues to melt, the ice pack could, depending on the scenarios considered, shrink outside the pole, between it and Greenland. A direct crossing by the NPR could reduce the crossing time down to 13–17 days [6], which would then compete with the NSR. In the medium term, will the battle in the Arctic between the two possible passages (NSR and NWR) be as epic as the one fought in the previous millennium against the severe climatic conditions encountered by explorers seeking to connect the Atlantic Ocean and the Pacific Ocean by a northern route?

To meet the objective of keeping global warming below 2 °C in 2050, maritime shipping must globally reduce its emissions by 2.6% per year between 2020 and 2050 despite the increase in traffic [7]. Using the NSR without any problems, in addition to saving time, also reduces fuel consumption by 40% and $CO_2$ emissions by at least 50% [8].

A northern shipping channel would be a competing itinerary to link Asian and European production and consumption centers. It will compete with another alternative to the current routes along India and either via the Suez Canal or around Africa: the new Silk routes, developed by China. Today, land transport is rare, rail transport between China, Korea and Europe represents only less than 4% of the total volume in 2017 [9].

This article will focus solely on the NSR by making a proposal to reduce the environmental impact of heavy vessel traffic in the Arctic. Indeed, as Vladimir Putin said in front of the Federal Assembly in 2018: "*The Northern Sea Route will be the key to developing the Russian Arctic and Far East*". While Didenko et al. [9] considers that the use of modern sea-river vessels combined with satellite-assisted navigation makes it possible to meet demand and, in particular, recent political decisions by the Russian Government, it is nevertheless necessary to study ways of limiting the environmental impact of major traffic in the Arctic Circle. Among the international points not yet arbitrated is the location of transport platforms.

This paper is structured as follows: after having described the history of the conquest of a maritime passage off the Russian coast, it presents in a second part the use that is currently made of it and the constraints related to its exploitation. Then, it discusses how traffic on the NSR could evolve before, in a fourth part, proposing a possible use of renewable energy resources, by matching the potential for energy production with the needs for a fleet of electric ships. Finally, a comparison with current circuits is presented, as well as a possible route for the NSR associated with a feasible establishment of different stopover ports.

*1.2. The Northern Sea Route in History*

In modern History, in the 11th century, Uleb, military chief of Novgorod, was the first to travel on the White Sea, without knowing the route to follow, the climatic conditions, the winds, the marine

currents and the precise location of the underwater lands [10]. The term NSR officially refers to the shipping channels between the Novaya Zemlya and the Bering Strait, located in Russian territorial waters [11]. By extension, here is also considered the part located off Norway as globally integrated into the NSR.

As early as the 16th century, it was proposed to link China and Europe by a more direct route to the North [10]. Navigation was carried out only in sections on the NSR. For the far East of its route, in 1648, 80 years before Bering, Yvan Ygnatyev was the first to succeed in connecting the Kolyma River to the Bering Strait. Before that, at the end of the 16th century, Willem Barents died in Novaya Zemlya while trying to sail on the NSR. This attempt marked the end of a period during which the search for a permanent passage was conducted, except for the Russians who claimed ownership of the lands and seas of northern Siberia and who remained the only ones to explore these icy territories. To this end, they developed the port of Arkhangelsk as a commercial and shipping port, linking the Dvina estuary and the West via the White Sea [10].

Then, in the 17th century, the conquest of the North of the current Federation took place from the great rivers and along the coast, gradually eastwards [12]. The first really accurate odds map was produced in 1763 by Lomonosov. He suggested a route that was attempted in 1765, 1768 and 1781, without success. The climate of the 17th century was warmer than that of the 18th century due to the decrease in global temperature following volcanic explosions, but less so than the 21st century. During this cold period, ships were systematically trapped in ice when trying to cross from the North, as in the case of the Chichagov expeditions in 1765 and 1766, which started from Arkhangelsk in the West on the White Sea, usually navigable in summer [10].

The true first complete crossing of a ship from the West (from Karlskrona, Sweden) to the East by the NSR, without damage caused on the ship by ice, was only carried out in 1878 and 1879 by the Finnish-Swedish scientist Nils Adolf Erik Nordenskjöld, along the coast. Living conditions in Arctic are not considered to be lenient [13]. The Arctic winter froze his ship near the Bering Strait, forcing him to complete his journey only the following summer. However, on the strength of this experience, he considered this passage through the NSR to be of no real economic interest because of the amount of ice encountered [14]. This attempt to cross the river was only repeated in 1915 and 1916, also requiring two summers. It was not until 1932 that the crossing was carried out in only one summer [10].

On the other hand, the NSR has been regularly exploited in its western part since the second half of the 19th century. Shipping remained weak but regular between Europe and the Kara Sea, mainly exporting minerals extracted from Siberia. Then, during the Soviet period, to ensure the economic development of the northern Russia, freight transport to the Ob and Yenisey estuaries is developed, although most freight transport to the western Russia was carried out on the two rivers [10]. A Soviet administration was created to develop the NSR, between the White Sea and the Bering Strait, both by sea and by air in 1932 [15]. As a result, about 100 polar stations, ports and roads have been built as a basis for development.

Under the aegis of this organization, a first icebreaker linked East to West in 1934. To make the crossing more operational and efficient, the basic idea of distributing resources throughout the route is deployed. Icebreakers are distributed throughout the route, and throughout separate NSR segments. Since then, nearly 250,000 tons of freight have been carried by sea since 1935, double the amount carried by rivers. The West of the NSR was still being exploited despite a disaster in late summer 1937, when a ship was trapped in the ice and drifted West for 812 days. Indeed, one of the consequences of the Coriolis effect is that the ice drifts from the pole westward, taking with it the ships that remain trapped there. Before World War II, more than 4 million tons of freight were transported.

*1.3. The NSR in Modern Times*

At the beginning of the last century, in Russian official documents, the NSR was defined as '*a national transport communication route historically developed by Russia*'. The periods of seaworthiness in the 20th century varied between 30 and 110 days. In 1955, the modeling of the ship motion in ice, including

the forces exerted on the hull by ice, was carried out. This improved and secured the navigation conditions. Then, tactics to allow a ship to sail alone or in convoy were defined. They integrate the speed of the vessel(s), density, pressure, ice concentration and the presence of hummock and ridge ice when the ice has been formed for several years. Then, in the early 1960s, icebreakers were modernized. The *Lenin*, the first nuclear-powered vessel, deployed 44,000 horsepower. In the 1970s, other nuclear icebreakers were move along the Yenisey River, opening a 9-m wide passage. More powerful, they almost allowed navigation on rivers all year round [10].

In 1991, a French National Marina ship, *the Astrolabe*, was the first non-Soviet ship to use the NSR without the help of icebreakers, crossing the tipping point around the Ob estuary [16]. Indeed, it is an intermediate point on the NSR between the operated European and Asian sides, still experiencing ice-related blockages in winter. To compete with the passage between the two continents using the Suez Canal, it is still necessary today to open the way with icebreakers, to travel on the Asian side. At first glance, the NSR can be considered financially attractive if the crossing is feasible at an icebreaker cost comparable to the amount of the Suez Canal toll. N. Otsuka [17] determined that using 20,000 tons vessels would be a form of optimum between capacity and short travel time.

On the NSR, up to 6 million tons were transported per year before the collapse of the Soviet Union. As a result, attendance declined to only 1.5 million tons in 1998, a level comparable to that of 1965 [10]. In 2009, international maritime transport was authorized on the NSR. Three years later, Gazprom chartered a first liquefied natural gas (LNG) carrier that left the Ob River estuary to travel the entire NSR [5].

Currently, the Federal Agency for Maritime and River Transport of the Russian Federation Ministry of Transport, defines the NSR route as the area between continental coast and the red line in Figure 1 [18]. A state agency is responsible for the development of this seaway, the Northern Sea Route Administration (NSRA). The NSRA has published that in 2018, the volume of freight traffic almost doubled in one year to 19.7 million tons (10.7 million tons in 2017). In detail, the freight was constituted as summarized in Table 1 [19]. The 11 million tons of freight that traveled on the NSR should be compared with the more than 1000 million tons that passed through the Suez Canal [2], which shows that there is a very significant growth potential for shipping on the Northern Route. The freight volume for 2019 is estimated at 26 million tons. For the following years, projections show a continuous increase in tonnages (Table 2).

**Table 1.** Composition of cargo carried in 2018 on NSR [19].

| Freight | General Cargo | Coal | Ores | Oil and Oil Products | Gas | Liquefied Gas (LNG) |
|---|---|---|---|---|---|---|
| thousand tons | 2340 | 291 | 43 | 7,810 | 805 | 8399 |
| change for 2017 | −6.3% | −16% | +30% | +16% | +750% | +3770% |

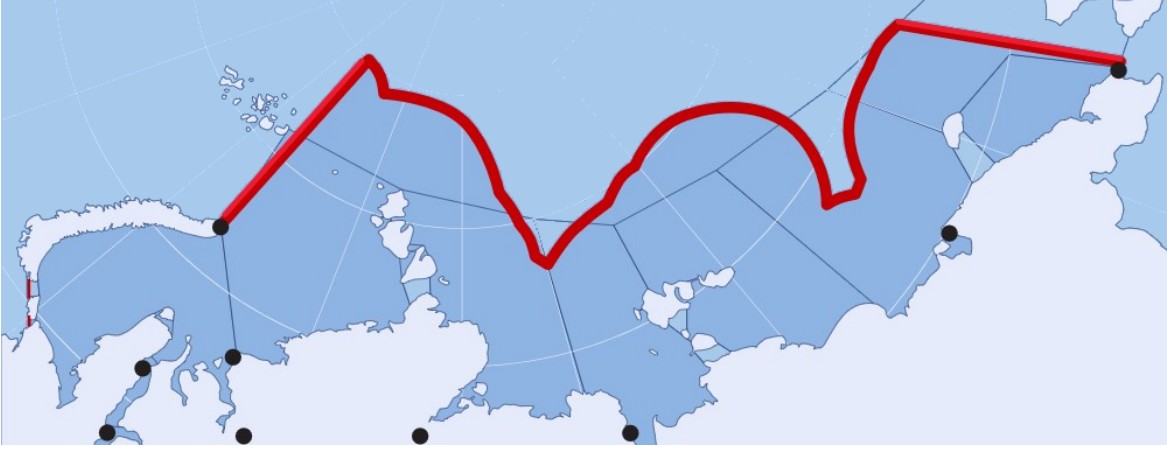

**Figure 1.** The water area of the Northern Sea Route [18].

**Table 2.** Projection of cargo quantity on the NSR.

| Year | 2017 | 2018 | 2019 | 2020 | 2021 | 2024 |
|---|---|---|---|---|---|---|
| million tons | 10.7 | 19.7 | 26 | 44 | 51 | 80 |

## 2. The Uses of the NSR

The Northern Sea Route has competitive advantages over the routes that must bypass India.

### 2.1. Market Situation

The NSR is the shortest route between Asia and Europe. It is on average 35% shorter than the route using the Suez Canal [20]. In reality, depending on the ports of origin and destination, the distance is less than 10 to 40%. Fuel consumption between Europe and Japan is 20% lower by using the NSR [2]. Table 3 shows the distances in kilometers between some European and Asian ports, considering four possible routes: the NSR, through the Suez Canal, around Cape Town and through the Panama Canal. The distance taken as a reference is that of the route passing through Egypt. This table shows that, except for the ports of South-East Asia, passage through the NSR is always the shortest route. According to [21] the sea route between Europe and China via the NSR requires 25 days and 625 tons of fuel oil and the passage through the Suez Canal, 35 days and 875 tons, representing a saving in time and fuel of nearly 30%.

**Table 3.** Comparison of some routes according to four sea routes.

| European Port | Asian Port | NSR | Suez | Cape Town | Panama |
|---|---|---|---|---|---|
| Hamburg | Hong Kong | 15,870 | 8800 | 25,970 | 27,150 |
|  |  | −11% | 0 | +40% | +38% |
| Rotterdam | Yokohama | 12,880 | 21,130 | 28,110 | 23,380 |
|  |  | −37% | 0 | +30% | +11% |
| Rotterdam | Shanghai | 14,420 | 20,010 | 26,850 | 25,250 |
|  |  | −24% | 0 | +31% | +26% |
| Rotterdam | HoChiMin City | 17,220 | 17,130 | 24,210 | 27,550 |
|  |  | +6% | 0 | +38% | +60% |
| St Petersbourg | Seoul | 15,720 | 23,310 | 31,500 | 28,120 |
|  |  | −33% | 0 | +35% | +21% |

The average navigation time along the NSR along the Siberian coast decreased from 20 to 11 days between 1990 and 2012 due to the reduction in ice [6]. Currently, some 50 ports mark the NSR coasts, the most important being, from West to East, Arkhangelsk (marked with an A on the map in Figure 2), Sabetta (S), Dikson (K), Dudinka (D), Igarka (I), Tiksi (T), Pevek (P) and Providzeniya (R). A first outline sketch passing near the coasts can then be established (in pink dotted lines on the map).

Today, the NSR is used mainly for the export of oil, LNG and wood and to import food into the coastal regions of Siberia. However, while this production is expected to increase until 2040, it is also expected to decrease rapidly and reach current production levels around 2050 [22]. Thus, the export of fossil resources alone does not seem to justify the development of maritime transport infrastructure along its route, although Aksenov et al. [6] believes that unescorted shipping should be possible as early as the 2030's. It is almost certain beyond 2050. However, even then, in winter, it will still be necessary for convoys to be accompanied by icebreakers. Currently, four nuclear icebreakers operate on this seaway [2]. Ships operating on the NSR are equipped with especially reinforced hulls. In addition, ice movements cause unpredictability of ship arrival times. N. Otsuka [17] proposes, since the cost is on average comparable, to use the NSR in summer and the Suez Canal in winter. Currently, this combination NSR/Suez has the same cost for a standard ship of 8000 tons. However, using the NSR would not be profitable with large tonnage vessels, especially since the shallow waters on this

route limit the size of the vessels [23]. Finally, Otsuka recommends that the NSR should only be used for freight requiring rapid delivery.

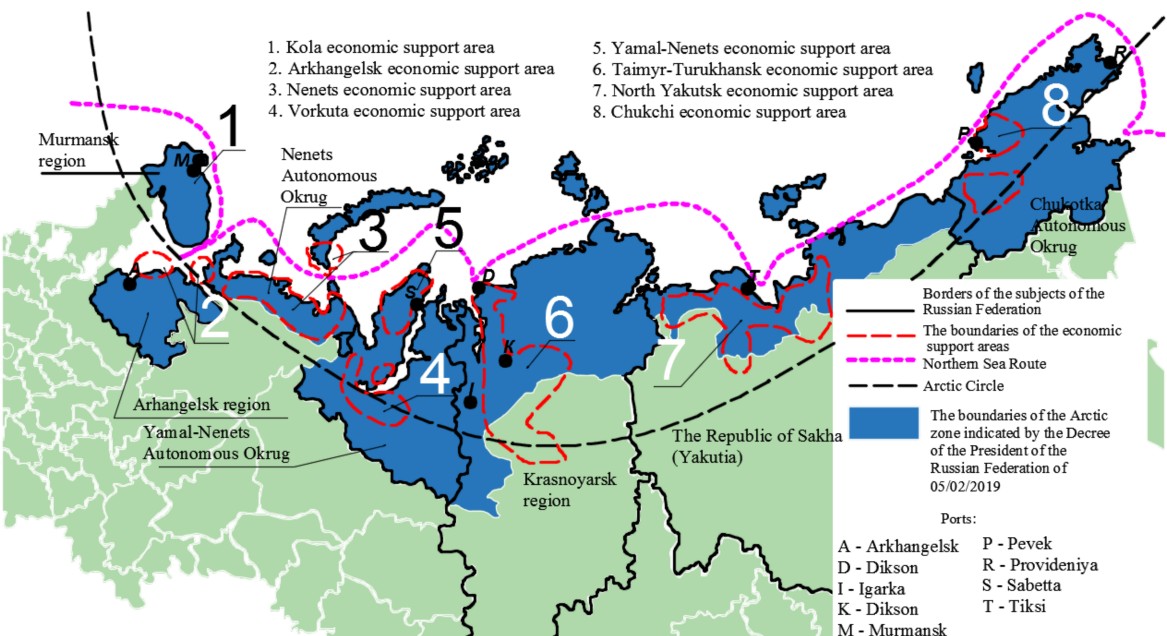

**Figure 2.** Major ports and economic support areas in the North of the Russian Federation.

## 2.2. Constraints And Limitations

Unlike the Soviet era, when the NSR was considered to allow the transport of large quantities of resources, with a view to increasing geopolitical and technical prestige, today the development of the NSR aims to develop international transit traffic by this northern route [1]. But not only that. The development of traffic on the NSR also has a geopolitical dimension, influencing global international processes by changing trade routes. Travkina et al. [1] think that five points are to be developed to facilitate the development of the NSR: the unification of the transport system by establishing fixed routes; the modernization and development of the Arctic fleet and port infrastructure; the provision of economic incentives; the increase in international freight transit; the modernization of container transport systems (loading, unloading, cargo storage in ports).

A major risk to shipping in the Arctic lies in shipping accidents causing oil spills [6]. To reduce the risks, it is necessary to plan the itinerary in advance [5]. There are different methods for dynamically optimizing ship navigation routes along the NSR, taking into account areas occupied by ice [24,25]. Avoiding ice is all the more crucial as the power deployed is obviously greater if a vessel has to cross an ice-covered sea and break ice sheets. The speed of ships traveling on the NSR, measured in 2014, is generally between 4.5 and 8 kt (8 and 15 km/h) throughout the route. Difficulties related to weather conditions, involving navigation stops, occur mainly along the entire coast of the Krasnoyarsk region and between the mouth of the Lena River and the new Siberian Island of the Novosibirsk Oblast [20]. This is particularly visible on the ice chart in summer [26]. On the other hand, even in winter, with the help of nuclear icebreakers, it is possible to travel at an average of 5 kt (almost 10 km/h) between Europe and Sabetta [17].

Baginova, Lyovin and Ushakov [27] proposes to measure the efficiency of an intercontinental transport system against five criteria: speed in goods delivery, shipment regularity, quality and speed of the handling of cargo at departure and arrival points all along the course, safety of goods during travel and the possibility of batch deliveries. To this should be added the cost of transport and seasonality: high cost dependence on the ability to navigate between ice in autumn and winter. According to these criteria, the recurrent presence of ice on the NSR is a major obstacle to its development.

Thus, the two major risks intrinsically linked to intensive use of the NSR are as follows:

- Risk related to the drift of ships that would remain trapped in the ice pack;
- Risk of pollution related to freight or fuel oil.

The first risk will decrease with the retreat of the ice. As already mentioned, it will always be necessary, in winter, to accompany ships with icebreakers. To reduce the second risk, it will be necessary to ensure that freight is not polluting and to eliminate the risk associated with fuel oil by using an electric propulsion system.

## 3. Development of Traffic on the NSR

### 3.1. Estimation of Energy Needs

To ensure that the northern territories of the Federation benefit from real economic spin-offs that contribute to the development of the territory, the development of the NSR must be coordinated with the development of land transportation infrastructures and transportation hubs [2]. Since land-based infrastructures allow for the efficient circulation of resources, they allow for the socioeconomic development of the territories. This is also true for the northern coasts of the Federation, or the ports of Dudinka (D) and Igarka (I), which are now connected by land and sea.

Most of the current development projects revolve around the exploitation and export of fossil resources. The largest deposits of the Arctic are located on the Yamal Peninsula. The total number of deposits is 32. All the Yamal Peninsula deposits reserves and resources are 300 million tons of oil, 1.6 billion tonnes of gas condensate and 26.5 trillion cubic meters of gas. The most important of these are Bovanenkovskoye. In this place, annual production is estimated at four million tonnes of stable condensate and 217 billion cubic meters of gas. Other sites also have significant potential: Novoportovskoye (oil and gas condensate reserves of 188.9 million tons of oil equivalent) [28], Zapolyarnoye oil and gas condensate (130 billion $m^3$ of gas per year, 80 million tons of gas condensate and oil), South Tambey gas field - Yamal-LNG project (reserves of 926 billion cubic meters of gas) [29]. The only field being developed on the Arctic shelf is Prirazlomnoye (reserves of 70 million tons of oil) [30]. The Russian Federation has defined its strategy for the development of the Arctic zone in 2017. It has divided the coastal regions into eight economic support zones from West to East: Kola (1, Murmansk region), Arkhangelsk (2), Nenets (3), Vorkuta (4), Yamalo-Nenets (5), Taimyr-Turukhansk (6, Krasnoyarsk territory), North Yakutsk (7) and Chukotka (8) [31], as shown in Figure 2.

To simplify the problem, let us consider in a first approach that a 10,000-ton vessel must have an engine power of 1 GW to move at a sufficient speed. A first approach to electrifying ships would be to place wind turbines to supply all or part of its energy needs. For example, three Flettner wind turbines on board could provide between 200 kW and 400 kW, according to Traut, Gilbert, Walsh, et al. [32]. This would provide about half the energy needed to propel a 5500-ton ship. However, the partly self-propelled ship would still need fossil fuel to make the crossing in one piece.

It therefore appears that, if the vessels are fully electric-powered, it will be necessary to make calls along the route. They will then have to either recharge their batteries or change them, through a swapping effect. This second solution is quicker, allows off-line battery maintenance operations and guarantees greater operational safety, as the on-board batteries are then checked before being put back into service. In order to harmonize the swapping conditions, it will be necessary to entrust the management of the storage and transfer infrastructures to a single company or consortium. The current Russian agency may be able to supervise this and ensure that management is consistent with its own objectives. The question of the ships and batteries ownership and the initial investment is not treated in this article because it remains the responsibility of the transport companies and the owners. A study to find out what is the break-even point in the economic model of battery swapping should be carried out to refine the project.

Maersk now uses 400-m long EEE vessels with two diesel engines developing between 30 and 68 MW. As seen above, this type of vessel is not suitable for operation on the NSR. We will therefore

study two types of vessels that can operate on the NSR: a 180,000-tonne bulk carrier, which would represent the maximum ship size, and a smaller freighter, carrying 800 evp (twenty-foot equivalent containers, a conventional unit for assessing the capacity of a transcontinental vessel).

## 3.2. Study Case

The 180,000-tons bulk carrier studied here is a ship that covers an average of 167,800 km per year (90,600 nautical miles per year, with 1 nautical mile is 1.852 km), consuming per year 13,700 tons of fuel oil, with an average sea speed of 27 km/h (14.5 kt) [33] with 1 kt is 1.852 km/h). Its engine develops an average power of 18.6 MW, consuming an average of 0.082 T/km (0.151 T/mile) of fossil fuel Table 4.

For its part, the small 800 evp container ship carries a total of 10,000 tons of cargo over 133,900 km a year. It consumes 7800 tons of fuel oil for an average engine power of 5.4 MW. It thus consumes 0.058 tons per kilometer. By considering the annual distance traveled and the average speed, it is possible to estimate the daily consumption at sea, at 53 and 39 tonnes of fuel a day respectively for the bulk carrier and the container ship. Usually, it is considered that a ship on intercontinental routes consumes a few tens of tons per day, depending on the vessel type, its tonnage and speed, as already written in part 2.1 (25 tons a day).

An intermediate size container carrier of 20,000 tonnes, as mentioned in part 1.3 as the optimum size to navigate on the NSR, whose engine develops a power of 13.2 MW, travels on average 118,500 km per year at an average speed at sea of 29 km/h, consumes for that 12,400 tons of fuel.

**Table 4.** Study datas, recalculated from [33].

| Ship | Bulk Carrier | Container Carrier (Estimated) | Container Carrier |
|---|---|---|---|
| capacity (evp) | 14,500 | 1600 | 800 |
| capacity (T) | 180,000 | 20,000 | 10,000 |
| annual average mileage (km) | 167,800 | 118,500 | 133,900 |
| average speed (km/H) | 27 | 29 | 28 |
| fuel oil annual consumption (T) | 13,700 | 12,400 | 7800 |
| engine power (MW) | 18.6 | 13.2 | 5.4 |
| average concumption (T/km) | 0.082 | 0.105 | 0.058 |
| navigation day consumption (T/day) | 53 | 73 | 39 |
| $CO_2$ emisions (T $CO_2$eq/year) | 41,000 | 37,400 | 24,800 |

After this quick inventory of the ship fuel consumption, the next point examine how electrical energy could replace fossil fuels for their propulsion. An initial feasibility calculation will determine whether it is realistic, with current technologies, to carry enough lithium batteries to ensure a point-to-point journey. Let us consider that each swapping port is 1000 km away and that ships manage to travel over the Arctic as over other seas, at the average speeds observed (respectively 28 and 27 km/h (15 and 14.5 kt) for a container ship of 800 evp and a bulk carrier of 13,500 evp), since we assume that the ice has retreated sufficiently. Thus, the ships have to sail continuously for periods of 36 and 37 h. With the current least efficient batteries (mass and energy density of 0.1 KWh/kg and 0.2 KWh/L), the batteries represent a mass of 1943 tons and a volume of 972 m$^3$ for the container ship, i.e., 25 evp, and 6926 tons for 3463 m$^3$, i.e., 90 evp, for the bulk carrier. This represents respectively 20% of the weight and 3% of the transportable volume for the former and 4% and 0.7% for the latter. It should be noted that the most efficient LiFePo4 batteries currently on the market have densities of 0.25 KWh/kg and 0.62 KWh/L respectively. Current research to improve battery performance focuses on the one hand on the nature and shape of the electrodes and on the other hand on the conductivity ionic of the electrolyte and the separator, for a larger storage capacity, more stability and larger operating temperature ranges [34]. In particular, the electrolyte can be liquid or solid. If the current liquid electrolyte offers densities of 0.250 KWh/kg, future liquid electrolyte lithium-ion batteries with metallic lithium should in some time commonly offer densities of 0.475 KWh/kg (0.400 KWh/kg without metal). The future solid state electrolyte batteries are announced at 0.480 KWh/kg [35].

To this, it is necessary to weight the estimate according to various parameters such as the technological evolution of the batteries, their aging, the need to cool the batteries and therefore the real necessary volume for the batteries and their control devices, the depth of discharge (DoD) of the batteries [36] and the imponderables linked to navigation. Respectively, in this first approach, we estimate each parameter as indicated in Table 5, which makes it possible to round off between 2 and 4 the correction coefficient to be applied to the previous results. Batteries are used until reaching a level of wear similar to that taken into account for an electric vehicle, i.e., up to a state of health (SoH) of 0.8, before being changed and recycled [36]. This leads to the summary of the requirements given in Table 6, for a swapping of 1000 km between ports-stages.

Compared to other lithium-ion battery technologies, LiFePO4 batteries can occasionally supply more power and are rechargeable at higher currents, therefore faster. They can be recharged a greater number of times, without the need to favor partial discharges. Their discharge voltage is more stable and they present a lower fire risk. The technology is mature and has an average cost of between 6400 and 13,600€ /ton, which places it among the cheapest on the market for current lithium batteries [34].

**Table 5.** Weighting parameters.

| Technology | Aging | Volume | DoD | Imponderable | Weight Coef. |
|---|---|---|---|---|---|
| 0.5 to 0.25 | 1.25 (SoH = 0.8) | 2 | 2 | 1.5 | 1.875 to 3.75 |

**Table 6.** Batteries required for each type of vessel.

| Ship | Energy per 1000 km | Weight | Volume | Evp Equivalents |
|---|---|---|---|---|
| container carrier 800 evp | 400–800 MWh | 4000–8000 T | 2000–4000 m$^3$ | 50–100 (6–13%) |
| bulk carriers 14,500 evp | 1400–2800 MWh | 14,000–28,000 T | 7000–14,000 m$^3$ | 180–360 (1–2.5%) |

It can thus be estimated that a 20,000-ton (1600 evp) ship would require between 625 and 1250 MWh to travel 1000 km at sea and occupy the volume of 80 to 160 evp.

## 4. Possible Use of Renewable Energy Resources

Renewable energy production is booming in the Russian Federation [37]. Now that the needs have been assessed, the feasibility of generating the electrical power needed to recharge the batteries moved from one port to another by ships must be verified.

### 4.1. Location of Energy Banks For Swapping

Kostin et al. [38] assessed how much energy can be obtained from wind power and solar radiation in the North of the Russian Federation. Savard et al. [39] mapped the maximum cumulative annual energy production potential per hectare (Figure 3) for the North of the Russian Federation. The map also shows in dark green the ice limits recorded in recent years in summer in the Arctic Sea [40].

To ensure the feasibility of producing this energy on site, because of the growing importance of autonomous electrical production in Arctic [41], it is necessary to integrate an additional constraint, that of the presence of natural parks protecting the land, particularly on Wrangel Island [42]. This eliminates some territories with the greatest potential. In the case of Wrangel Island, if it could not be used as a port of call, it would be possible to use the Leningradsky area, located at almost the same longitude, but on the mainland, since the potential for renewable electric power generation is equally high there. A new route is possible for the NSR, based on the route already proposed, taking into account environmental constraints, the potential for renewable energy production and the location of ports. It is drawn in dark red in Figure 4.

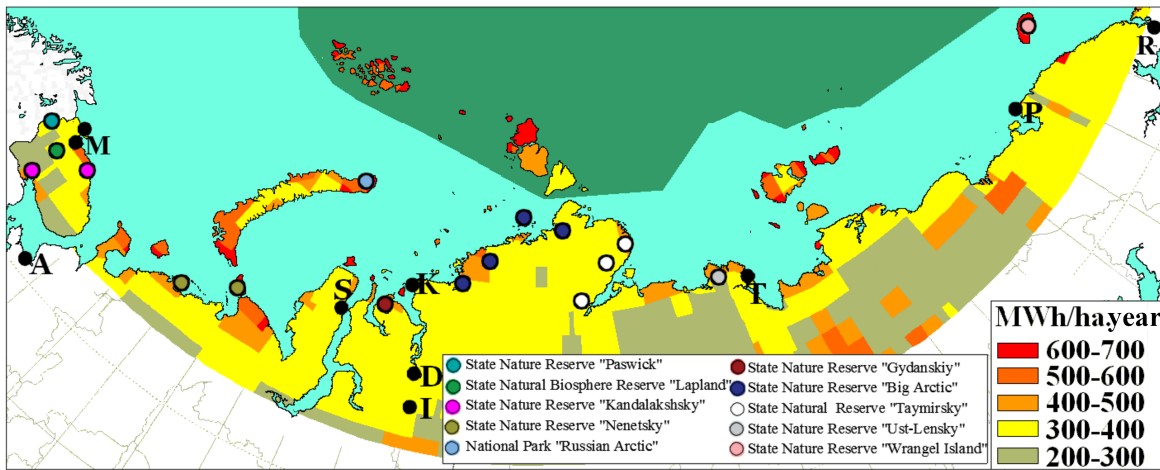

**Figure 3.** Map illustrating the average potential for electricity generation in mega wattheure from renewable sources per hectare and per year [39], the main Arctics ports and the natural parks.

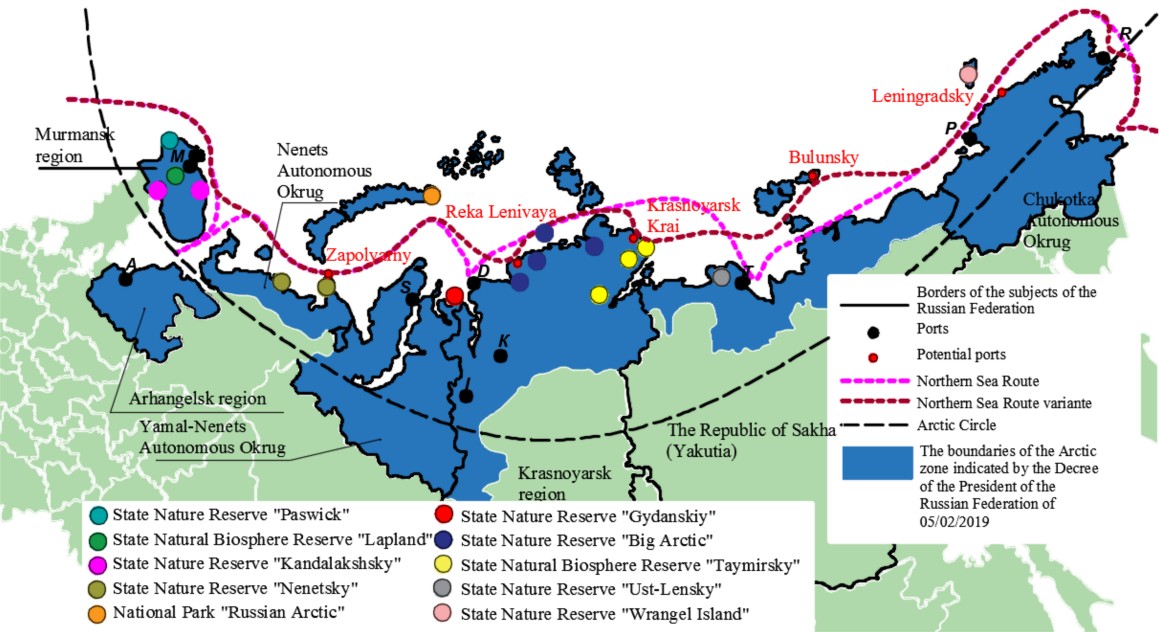

**Figure 4.** A possible route for one NSR per port of call.

The second aspect to be determined is the spacing between each port-stage. In the framework of this study, we will retain a distance between reloading harbors of around 1000 km. By considering only the ports located on the Arctic shores of the Federation, we will consider those located in the areas with the greatest renewable energy potential, outside the protected areas. They are listed in Table 7, which also shows the distance between them. Beyond that, towards the West, to travel between Rotterdam and Shanghai for example, the same type of equipment should be added, located at similar spacing, namely around Moskenes Oya in Norway and close to the Shetland Islands in Scotland. In this latter case, to preserve the nature reserve, an off-shore port, powered by wind turbines and tidal turbines in the North Sea Doggerland, should be set up. Similarly, towards the East, stopovers should be added in Provideniya, Pakhachi, South Kamchatka, South Sakhalin Island, Vladivostok, before reaching the main Asian ports.

**Table 7.** A possible location of energy storage banks on the northern shores of the Russian Federation.

| Location | Leningradsky | Bulunski South District | Krasnoyarski Kray | Reka Lenyvaya | Zapolyarny | Murmank |
|---|---|---|---|---|---|---|
| Distance to the next port | 1160 km | 1050 km | 850 km | 1140 km | 950 km | - |

In total, between Rotterdam and Shanghai, there are 13 ports of call that should be arranged along the route, plus those of departure and arrival in Asia and Europe.

### 4.2. Estimated Needs for a Fleet of About 25 Vessels

Ships of 20,000 tons require between 625 and 1250 MWh to travel 1000 km. Taking into account the maximum spacing between ports of call, it is, therefore, necessary to produce, in the worst case, 1500 MWh per ship at each port of call. With an average production of 600 MWh/ha.year in areas affected by the presence of a port-ship, 1000 hectares would have to be devoted to supply a single ship in one day. In total, out of the 13 ports, an area of 41 by 41 km would then have to be used to supply one ship every day in each port. A Rotterdam-Shanghai trip made by the NSR at an average of 15 kt would require 23 days of navigation and 13 half-days of swapping, for a total of 36 days per trip. Taking into account the loading, unloading and maintenance operations of the ships, each of them could make 8 crossings in both directions each year. The swapping and navigation time between two ports is approximately two days. As there are 13 stages on the route, 13 ships can use the NSR in each direction. This would mean operating a fleet of 26 operational vessels, to which it is possible to add reserve vessels used to replace those involved in maintenance or heavy repairs. In one year, the NSR can potentially transports from one continent to another a total of 158,000 evp per direction and about 4 million tons of freight, the level transported in 2017, including the export of fossil fuels.

### 4.3. Economic Comparison of Operating Costs

To date, the production cost, including investments, of one MWh of renewable energy is estimated, for wind energy at 100€ and for solar energy at 150€ in France [43]. The Russian Association of Wind Power Industry gives less frightening figures, indicating that by 2030, the average production cost of wind MWh could drop to 21€ (in the United States) [44]. Its president, Igor Bryzgunov, specifies in [45] that the first onshore wind turbines were installed in 2013 in the federation. At the start of 2020, they had a production capacity of around 200 MW and 400 MW are under construction. For 2024, the objective of a production capacity of 3.4 GW was announced. The cost of the MWh of wind origin should then pass from 50€ in 2015 to 28€ in 2030. We therefore consider for the study that, taking into account the often adverse climatic conditions, the production cost of the MWh of renewable energy in the North of the Russian Federation can be globally close to 50€ whatever its production source. Thus, the energy produced for a 1600 evp container ship (i.e. a minimum of 1440 evp of useful capacity) consuming between 800 and 1600 MWh of electrical energy (Table 6) between two ports, costs to produce between 40,000 and 80,000€. This vessel would consume 0.105 T/km if it operated on fuel oil. For a journey at sea of 1000 km, it will then consume 105 tons of fuel oil. The price of heavy fuel oil is volatile. For example, on 30 November 2019, it varied from 250$ to 500$/T depending on its sales spot (specifically from Rotterdam to Los Angeles). On the Saint Petersburg market, heavy fuel is currently trading between 500 and 700€ while in the northern ports of the Russian Federation, such as Murmansk, the ton costs around 360€. With an average cost of 400$/T, or 350€/T, the cost of fuel oil is only close to 40,000€, i.e., on average 50% of the cost of producing the same electrical energy.

The cost of producing the electrical energy needed to fully electrify ships is, to date, more than five time the current cost of heavy fuel oil. Despite the environmental inconvenience of this viscous product, loaded with impurities and therefore numerous toxic substances, it remains cheaper in terms of marketing costs. This fuel contributes in particular to nitrogen oxides (NOx) and sulfur oxides (SOx) pollution. Moreover, it is commonly accepted that 12% of $CO_2$ emissions in the European Union come

from maritime traffic. If world oil and gas prices were to remain at their current levels, the cost of producing renewable energy would have to be halved or the amount of energy needed to move ships would have to be halved. Consider the possible solution presented in [32] of equipping ships with wind turbines to combine mechanical propulsion and wind power, but without sails. The average power produced in autonomy thanks to the wind is then, according to the power of the winds on the routes taken, at least 193 kW. During the 36 h of navigation, a ship equipped with a wind turbine with a Flettner rotor would produce an autonomous quantity of almost 7 MWh, or less than one percent of the energy required. This self-generation remains too marginal to reverse the economic situation. Improving the mechanical efficiency of ship engines is another avenue that should not lead to the necessary leap in competitiveness. Moreover, this total electrification solution could make economic sense only if the cost of fuel oil increases more than twice.

However, these calculations should be weighted according to the time saved and the lower consumption compared to current roads. Taking as an example only the trips between Rotterdam and Shanghai, which are 28% shorter than those via the Suez Canal (Table 3), the additional cost of fuel (electrical energy) is only 62%. The volatility of hydrocarbon prices is very high (the price of a barrel having evolved between 25$ and 135$ a barrel in seven years at the beginning of the century). The quality of hydrocarbons decreases with the duration of exploitation of this energy sector. With the beginning of the development of the Prirazlomnoye field on the Arctic shelf, a new grade of ARCO oil appeared/ARCO oil has a high density (about 910 kg per cubic meter), high sulfur content and low paraffin content. In the Arctic Novoportovskoye fields (YaNAO), the oil grade is called Novy Port, it belongs to the category of light with low sulfur content (about 0.1%) [28,30]. Moreover, it seems archaic to foresee a future with hydrocarbons as a source of energy for transport, when they should be reserved for more noble uses (chemical processing industry) and when there are so many less polluting and less dangerous sources of energy.

The International Maritime Organization (IMO) has set the societal objective of reducing $CO_2$ emissions for all maritime transport international by at least 40% by 2030, and by 70% by 2050, compared to 2008; and GHG (GreenHouse Gas) emissions by at least 50% by 2050 [46]. The heavy fuel oil used for maritime transport includes a large proportion of viscous residues, metal and sulfur. It is obtained at the end of refining petroleum in different types of fuels and before bitumens. Burning it releases a lot of NOx and SOx (nitrogen oxides and sulfur oxides). On average, according to figures from the Intergovernmental Panel on Climate Change (IPCC) [47], GHG emissions related to maritime freight transport are between 10 and 40 g $CO_2$eq/T.km. One gram of $CO_2$ equivalent ($CO_2$eq) corresponds to the mass of carbon dioxide product in a combustion, with the same global warming potential as any other greenhouse gas. Table 5 last row presents the quantity of $CO_2$eq released on average per tonne of freight and per kilometer for the three types of vessels mentioned [48]. If the 26 vessels of 20,000 tonnes were used to their maximum, for a route of 14,420 km (Table 4) between Rotterdam and Shanghai, this would be 623,000 tonnes of $CO_2$eq that would not be emitted each year. In fact, apart from construction, establishment and end of life, the sources of renewable energy production (wind and solar) do not emit $CO_2$ during their production period.

This first step in determining the relevance of developing an electric vessel fleet operating on the NSR should continue with more precise studies, including in particular the different speeds along the route as well as seasonal effects. Investment costs for on-board batteries should gradually decline as the technology sector matures. In addition, research on improving energy density by mass and volume will lead to a reduction in the size and volume of electrical energy storage systems and therefore to a reduction in investment costs. To achieve the objectives of the IMO, other solutions than the electrification of ships are being studied, such as the use of Liquid Natural Gas [49]. As the wind and solar power sector matures, unit costs will decrease. For its part, over long periods, the price of hydrocarbons should not decrease. Thus, in a few years, the propulsion of ships by electric energy will cost less than the use of heavy fuel. It is therefore an opportune time to think about a change in technology.

## 5. Conclusions

This paper discusses the feasibility of fully electrifying the ships that will be sailing the NSR in the near future. Indeed, this new seaway will develop due to global warming as a result of the reduction of distances—and therefore transport costs—compared to other routes currently operated between East and West. In order to reduce the environmental risks associated with shipping, it will still be necessary to accompany ships by icebreakers in winter, to ensure that freight is not polluting and to eliminate the risk associated with fuel oil by using an electric propulsion system. A possible technical solution is to produce electrical energy stored in high-capacity batteries in ports of call about every thousand kilometers along the NSR. Ships on the route would then call in to change their discharged batteries for fully operational batteries.

The Russian Federation has a strategic card to play in order to preserve the extremely fragile ecosystem of these northern coasts by requiring that ships using this new maritime route do not pollute the air or the environment in the event of damage. This can be achieved by using a fleet of electrified ships with a tonnage adapted to the shallow waters of the Arctic. However, although the ecological interest is obvious, the fact remains that with the current solutions for both the production and storage of electrical energy, the use of heavy fuel oil powered shipping vessels is still economically advantageous. In the first instance, as with motor vehicles, it will be necessary to go through a stage of hybridization of the ships engines. It is now possible to consider and plan a future change in the propulsion methods of ships that will use the NSR.

To be able to organize totally clean traffic on the NSR, an organization, national or international, building on existing structures or not, will have to be set up with the task of managing ship traffic, battery swapping and possibly energy production and port management. This strong political affirmation of a truly sustainable development will, moreover, make it possible to develop economically the regions close to the ports along the Arctic Ocean and the Bering Sea.

**Author Contributions:** C.S.: Conceptualization, Writing, Review, Methodology; A.N.: Methodology, Resources, Writing, Review, Cartography; C.M.: Writing, Review, Translation; E.M.: Cartography. All authors have read and agreed to the published version of the manuscript.

**Funding:** The part of this research fulfilled in Saint-Petersburg Mining University was funded by the Russian Science Foundation, project No. 17-78-20145.

**Conflicts of Interest:** The authors declare no conflict of interest.

## Abbreviations

The following abbreviations are used in this manuscript:

| | |
|---|---|
| NSR | Northern Sea Route |
| NSRA | Northern Sea Route Administration |
| IMO | International Maritime Organisation |
| IPCC | Intergovernmental Panel on Climat Change |
| LNG | liquefied natural gas |
| MW | unit of power, one million of Watt |
| GW | unit of power, one billion of Watt |
| KWh | unit of energy, one hundred Watt in one hour |
| MWh | unit of energy, one million Watt in one hour |
| KWh/kg | mass energy density, one hundred Watt in one hour in one kilogram material |
| KWh/L | volume energy density, one hundred Watt in one hour in one liter material |
| MWh/ha.year | produced energy (in MWh) in one year on one hectare |
| $CO_2$eq | $CO_2$ equivalent: mass of carbon dioxide product in a combustion |
| $gCO_2$eq/T.km | $CO_2$eq product for one ton freight transported over one kilometer |
| GHG | GrennHouse gas |
| $/T | price in Dollars for on ton on material |
| € /T | price in Euro for on ton on material |
| A | Arkhangelsk port |
| D | Dudinka port |
| I | Igarka port |

| K | Dikson port |
|---|---|
| P | Pevek port |
| R | Provideniya port |
| S | Sabetta port |
| T | Tiksi port |
| evp | twenty-foot equivalent container |
| DoD | battery deep-of-discharge |
| SoH | battery state-of-health |

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
