# Peer review of "The Electrification of Ships Using the Northern Sea Route: An Approach"

_2199-8531, doi:10.3390/joitmc6010013_

Round 1

Reviewer 1 Report

Detailed remarks are described with highlighted text. Please change units of distance and speed unify. It means please use kilkometers and kilometers per hour or nautical miles and knots (nautical miles per hour). Also please use and correct data in the text for 1 nautical mile = 1852 metres, where it is necessary. Sub-chapters 1.1, 1.2 and 1.3 are not necessary in this paper. Should be reduced to half page only. Instead of it better to add sub-chapter about development of new batteries with conclusion why mentioned in the text LiFePo4 batteries were used in this paper for calculation.

Author Response

Consecutive corrections to your informative remarks are written in green in the new version of the manuscript.

Thanks for your beneficial comments. We have revised the corresponding errors in paper according to your good comments. We have integrated your remarks. If, however, the modifications made need to be completed, do not hesitate to let us know so that this paper can be qualified for this journal.

Detailed remarks are described with highlighted text. Please change units of distance and speed unify. It means please use kilkometers and kilometers per hour or nautical miles and knots (nautical miles per hour). Also please use and correct data in the text for 1 nautical mile = 1852 metres, where it is necessary.

Thanks for your helpful comment. We have taken up the remarks noted in the pdf below, indicating the pages and rows in the first version of the paper. During the first quotation, we indicated the data in kilometers and km/h as well as in nautical miles and knot. Then, we performed the demonstration using only the former. Indeed, it was a transcription error for the kilometer value of nautical miles.

Sub-chapters 1.1, 1.2 and 1.3 are not necessary in this paper. Should be reduced to half page only.

Thanks for your meaningful comment. It is true that this part devoted to history could be reduced. However, we suggest to keep it as it is, because it is part of the work carried out, in order to demonstrate that there has been a long-standing question of using the Arctic Sea to connect East and West. That's why, we add in part 1 introduction. "It has been a long-standing question of using the Arctic Sea to connect East and West".

Instead of it better to add sub-chapter about development of new batteries …

Thanks very much for your useful comment. We add a short paragraph on this point in part 3.2, to complete our article along the lines requested: "Current research to improve battery performance focuses on the one hand on the nature and shape of the electrodes and on the other hand on the conductivity ionic of the electrolyte and the separator, for a larger storage capacity, more stability and larger operating temperature ranges [Duan19]. In particular, the electrolyte can be liquid or solid. If the current liquid electrolyte offers densities of 0.250 KWh/kg, future liquid electrolyte lithium-ion batteries with metallic lithium should in some time commonly offer densities of 0.475 KWh/kg (0.400 KWh/kg without metal). The future solid state electrolyte batteries are announced at 0.480 KWh/kg [Ulvestad18].

[Ulvestad18] A. Ulvestad, A Brief Review of Current Lithium Ion Battery Technology and Potential Solid State Battery Technologies, available online https://arxiv.org/ftp/arxiv/papers/1803/1803.04317.pdf, 2018

[Duan19] J. Duan, X. Tang, H. Dai et al., Building Safe Lithium-Ion Batteries for Electric Vehicles: A Review. Electrochemical Energy Review (2019). https://doi.org/10.1007/s41918-019-00060-4, 2019

...with conclusion why mentioned in the text LiFePo4 batteries were used in this paper for calculation.

Thanks for your meaningful comment. Two paragraphs later, we add a justification for choosing this technology: “Compared to other lithium-ion battery technologies, LiFePO4 batteries can occasionally supply more power and are rechargeable at higher currents, therefore faster. They can be recharged a greater number of times, without the need to favor partial discharges. Their discharge voltage is more stable and they present a lower fire risk. The technology is mature and has an average cost of between 6,400 and 13,600 €/ton, which places it among the cheapest on the market for current lithium batteries [Duan19]. "

Page 2, row 49 : The text "from 13 to 17" may be understood as misleading sequency - reduced form 13 days now to 17 days in the future (illogical). I suggest follow text from original publication it means replace "from 13 to 17" with "to 13-17" or "down to 13-17"

Thanks for your valuable comment. Correction performed, in green.

Page 2, row 74 and page 6, row 203: « "Boats" mean very small floating devices, in practice of lenght below 25 meters. This article is about floating devices much longer then 25 meters. Then proper name is "ships" or "vessels". It should be used in text consequently. »

Thanks for your instructive comment. You're totally right. The term "boat" has been replaced in the text.

Page 3, rows 86,87,104,114,, :

Thanks for your precious comment. We corrected Bering's name.

Page 3, rows 112,113 :

Thanks very much for your constructive comment. We added the article in front of "northern" and "western Russia".

Page 4, row 150 : « the NSR is not proceeding along the red line. Theis red line is linit of the NSR from the north side. Then I suggest write instead "as area in between continental coast and the red line in Figure 1" »

Thanks very much for your constructive comment. Indeed, this is the northern end of the possible corridor for the NSR. We have corrected the text as proposed as well as colorized the area of the Arctic sea affected differently.

Page 5, row 160,

Thank you for your constructive comment. We add "Sea" between Northern and Route in the sentence.

Page 8, row 286 : « What miles??? Please use nautical miles. »

Thank you for pointing this out to us. We apologize for this error and correct miles in nautical miles, the calculations having been performed in nautical miles.

Page 8, row 587 : « First question. In western system dot separate full number/value from decimals of the value. It means 1 kilometer and 852 metres gives 1852 metres or 1,852 meters or 1.852 kilometers. Comma separate every three numbers. It means 1852 can be describes as 1,852. »

Thanks very much for your constructive comment. In English, a comma separate three numbers and a dot separate a decimal number, but in French, it is exactly the opposite. As a result, it happens to mix the two systems. Thank you for bringing this problem to our attention which, despite re-readings, only an outside expert eye can directly discern. We apologize for our low grammatical and syntactic level in English. We have read and corrected the errors we found. If you still notice problems that need to be corrected, please don't hesitate to report them for helping us to focus on translation errors.

« Second matter. One nautical mile is 1,852 meters = 1.852 km. I think 1,609 km is mistake. Instead of "1,609 km" should be "1.852 m" »

Thank you for your helpful comment. It is indeed one kilometer and 852 meters.

Page 8, row 587 : « It is not clear - it is consumption for 24 hours or 90600 miles. also please made all data in one system of units - kilkometers and kilometers per hour or nautical miles and knots (nautical miles per hour). »

Following your relevant remark, to help the reader, we add "per year" to specify the period of fuel consumption. We also corrected the text by leaving references to kt and miles only during the first example mentioned.

Page 8, row 273 : « line above is writtem 81 ton per day. Here is 100-300 tons per day. What is size of vessel consuming so much fuel???

For your guidance - standard PANAMAX vessel (70000 tons of cargo) consumes 25-32 tons per day. »

Thanks for your valuable comment. In order not to clutter up the reading, we simplified the sentence by: "it is considered that a ship on intercontinental routes consumes a few tens of tons per day, depending on the vessel type, its tonnage and speed, as already written in part 2.1 (25 tons a day)".

Page 9, figure 3 : « Description "MWh/ha.an" is not clear. Please write full description in the title of Figure 3. p.ex. "MWh/ha.an it is ....." »

Thanks for your precious comment. To clarify the legend, we added that the data were in mega watt hours per hectare per hour and corrected the French word "an" by "year".

Page 11, row 280 : « What do you mean "inter-distance between ports-stage" ???

Is this distance of approximately 1000 km in between consecutive ports of replacement of batteries? Please clarify in the text. »

Thanks for your meaningful comment. You are right, the sentence is not clear enough. This was to indicate that for about 1,000 km, the average consumption will be 228 tonnes. 1000 km is the proposed distance between two ports. We rewrote the sentence as follows: "This vessel would consume 0.105 T/km if it operated on fuel oil. For a journey at sea of 1,000 kilometers, it will then consume 105 tons of fuel oil."

Page 11, row 351 : « Please add information on prices being more related to the Northern Sea Route. Per example give prices for realistic supply port Rotterdam, Murmansk, Yokohama and Shanghai. »

We add, after describe the price of heavy fuel oil : “On the Saint Petersburg market, heavy fuel is currently trading between 500 and 700 €, while in the northern ports of the Russian Federation, such as Murmansk, the ton costs around 360 €.”

Page 11, row 380 : « I suggest replace "food" with "energy" »

Thanks very much for your useful comment. We change the word.

Page 12, row 411 : « To maintain order please add here also: symbol of Euro, symbol of USD Dollar, MHw, MWh/ha.year, MHw/ha.an, T, EEE, GW, MW, KWh/kg, KWh/l, KWh. It means, do not leave the Reader of this article to make his own interpretation of the symbols/abbreviations. »

Thanks for your valuable comment. We have added these abbreviations in the symbol table.

Reviewer 2 Report

Table 6. is not corrected (format) Content and review article. Many facts have been cited, but not all sources are publicly available. The article contains an aspect of logistics. There is no clear forecast, many facts are interesting. Present more comparative analysis referring to reliable and fundamental sources.

Author Response

Consecutive corrections to your informative remarks are written in red in the new version of the manuscript.

Thanks for your beneficial comments. We have revised the corresponding errors in paper according to your good comments. We have integrated your remarks. If, however, the modifications made need to be completed, do not hesitate to let us know so that this paper can be qualified for this journal.

Table 6. is not corrected (format)

Thanks for your meaningful comment. We have corrected the table format (new Table 7).

Content and review article. Many facts have been cited, but not all sources are publicly available.

Thanks for your precious comment. It is true that the reference [12] of 1937 is difficult to find. We have indicated the translation of the title in English, as well as the publishing house. Below is the full reference, in Russian and English:

P.K. Khmyznikov, П.К. Хмызников, Архипелаг Новосибирск Их остронов, Хатангский залив и усеь е реки Хатанги, Ленинград (Novosibirsk Archipelago of their islands, Khatanga Bay and the river Khatangi), in Материалы для лотереи моря Лаптевых и Восточно-Сибирского моря (Materials for the lottery of the Laptev Sea and the East Siberian Sea), Leningrad, Glavsevmorputi publishing house, 1937.

Some articles and documents are written in French or Russian but are available on the internet. Please, can you tell us which document(s) you did not obtain access to?

The article contains an aspect of logistics. There is no clear forecast, many facts are interesting. Present more comparative analysis referring to reliable and fundamental sources.

Thanks for your instructive comment. We did not focus our article on logistics and supply chain. Its ambition is not to propose one or more solutions for the commercial exploitation of the NSR but to evoke the feasibility of electrifying the ships which will use it in the relatively near future. Other articles, notably [Otsuka17 (19), Grigoryev17 (22), Farré14 (23), Kotovirta09(24), Pastusiak16 (25) ], deal specifically with some of these aspects.

Reviewer 3 Report

Interesting work that deserves publication subject to revision/consideration of the points below:

Section 3.2 needs substantial revision; the reader cannot follow the development of the case, therefore I strongly suggest to provide tables with all necessary data (input) and related calculations. In addition, the authors should provide a clear breakdown of the calculation related to the consumption of the diesel vs electric propulsion plant. The difference in the cost of energy is not fully clear. The same applies to the initial acquisition cost of the ships; see for example the concerns of Schinas et al. (doi:10.1016/j.oceaneng.2016.04.031 and doi:10.1016/B978-0-12-813830-4.00007-1) indicatively. It seems that the analysis does not consider the higher initial cost as well as the different configuration of the propulsion plant. Moreover, the above calculations should consider the environmental burden in terms of CO2eq as per IPCC (see Global Warming Potential, etc.) for both energy sources, i.e onboard and ashore. Otherwise the comparison is not fully fair. A port hopping schedule might also consider different speeds as well as risks, therefore the total environmental burden of the fleet, should also consider different speeds among arcs of the network as well as seasonal effects, i.e. regions that face higher risks of icing and therefore or restricted navigation, etc. 

Author Response

Consecutive corrections to your informative remarks are written in blue in the new version of the manuscript.

Thanks for your beneficial comments. We have revised the corresponding errors in paper according to your good comments. We have integrated your remarks. If, however, the modifications made need to be completed, do not hesitate to let us know so that this paper can be qualified for this journal.

Interesting work that deserves publication subject to revision/consideration of the points below:

Section 3.2 needs substantial revision; the reader cannot follow the development of the case, therefore I strongly suggest to provide tables with all necessary data (input) and related calculations.

Thanks for your valuable and precious comment. We add a table (Table 4) showing the data, deleted the values indicated which did not clarify the understanding and added in the text some details on the calculations made, in blue. For instance: “An intermediate size container carrier of 20,000 tonnes, as mentioned in part 1.3 as the optimum size to navigate on the NSR, whose engine develops a power of 13.2 MW, travels on average 118,500 km per year at an average speed at sea of 29 km/h, consumes for that 12,400 tons of fuel.

In addition, the authors should provide a clear breakdown of the calculation related to the consumption of the diesel vs electric propulsion plant.

Thanks for your helpful comment. To better mark the transition, we added: “After this quick inventory of the ship fuel consumption, the next point examines how electrical energy could replace fossil fuels for their propulsion. "

We have revised the statistics based on sources closer to the ground. This does not change the conclusions but is more in line with reality. We add, in part 4.3: “The Russian Association of Wind Power Industry gives less frightening figures, indicating that by 2030, the average production cost of wind MWh could drop to 21 € (in the United States) [1]. Its President, Igor Bryzgunov, specifies in [2] that the first onshore wind turbines have been installed since 2013 in the federation. At the start of 2020, they have a production capacity of around 200 MW and 400 MW are under construction. For 2024, the objective of a production capacity of 3.4 GW is announced. The cost of the MWh of wind origin should then pass from 50 € in 2015 to 28 € in 2030.” We therefore retain for this comparative study a value of 50 €/Mwh, based on more local and recent datas.

[1] : Russian Association of Wind Power Industry, Wind power costs: another 50 percent reduction possible by 2030, available online : https://rawi.ru/en/2017/08/wind-power-costs-another-50-percent-reduction-possible-by-2030/ , accessed on 05/02/20 ; 2017.

[2] : interviews – Russia - Wind energy in Russia: An interview with Igor Bryzgunov of the Russian Association of Wind Power Industry (RAWI), Renewable Energy Magazine, 22 octobre 2019, available online : https://www.renewableenergymagazine.com/interviews/wind-energy-in-russia-an-interview-with-20191022 , accessed on 05/02/2020, 2019.

The difference in the cost of energy is not fully clear. The same applies to the initial acquisition cost of the ships; see for example the concerns of Schinas et al. (doi:10.1016/j.oceaneng.2016.04.031 ...

Thanks for your meaningful comment. We add before the conclusion, after the detailed paragraph below concerning the carbon impact, the following sentence: "To achieve the objectives of the IMO, other solutions than the electrification of ships are being studied, such as the use of Liquid Natural Gas [Schinas16].

O. Schinas, M. Butler, Feasibility and commercial considerations of LNG-fueled ships, Ocean Engineering, vol. 122, pp. 84-96, 2016.

We also add, in perspective : “As the wind and solar power sector matures, unit costs will decrease. For its part, over long periods, the price of hydrocarbons should not decrease. Thus, in a few years, the propulsion of ships by electric energy will cost less than the use of heavy fuel. It is therefore an opportune time to think about a change in technology.”

...and doi:10.1016/B978-0-12-813830-4.00007-1) indicatively.

Thanks for your instructive comment. On this second point, concerning the acquisition strategies of the ships, the source of propulsion power appears only as a point in the set of criteria which contribute to the choice of the ship. Our article does not aim to offer a decision aid applied to the acquisition but to initiate reflection, this is why we did not go into details on this point. We add at the end of part 3.1 a sentence along these lines: “the question of the ships and the batteries ownership and the initial investment is not treated in this article because it remains the responsibility of the transport companies and the owners. A study to find out what is the break-even point in the economic model of battery swapping should be carried out to refine the project.”

It seems that the analysis does not consider the higher initial cost as well as the different configuration of the propulsion plant.

Thanks for your precious comment. Indeed, for the same reason, we have not included the investment costs as such.

Moreover, the above calculations should consider the environmental burden in terms of CO2eq as per IPCC (see Global Warming Potential, etc.) for both energy sources, i.e onboard and ashore.

Thanks for your instructive comment. We add the last row of new Table 4 in section 3.2 to specify the carbon impact.

In the economic comparison section, we have added a point on the reduction in GHG induced by this electrification project: “The International Maritime Organization (IMO) has set the societal objective of reducing CO2 emissions for all maritime transport international by at least 40% by 2030, and by 70% by 2050, compared to 2008; and GHG (Green House Gas) emissions by at least 50% by 2050 [3]. The heavy fuel oil used for maritime transport includes a large proportion of viscous residues, metal and sulfur. It is obtained at the end of refining petroleum in different types of fuels and before bitumens. Burning it releases a lot of NOx and SOx (nitrogen oxides and sulfur oxides). On average, according to figures from the Intergovernmental Panel on Climat Change (IPCC) [4], GHG emissions related to maritime freight transport are between 10 and 40 gCo2eq/T.km. One gram of CO2 equivalent (CO2eq) corresponds to the mass of carbon dioxide product in a combustion, with the same global warming potential as any other greenhouse gas. Table 4 last row presents the quantity of CO2eq released on average per tonne of freight and per kilometer for the three types of vessels mentioned [5]. If the 26 vessels of 20,000 tonnes were used to their maximum, for a route of 14,420 km [table 3] between Rotterdam and Shanghai, this would be 623,000 tonnes of CO2eq that would not be emitted each year. In fact, apart from construction, establishment and end of life, the sources of renewable energy production (wind and solar) do not emit CO2 during their production period. "

We add the last line of Table 4 in section 3.2 to specify the carbon impact.

[3] : International Maritime Organization. UN Body adopts climate change strategy for shipping, Available online: http://www.imo.org/en/mediacentre/pressbriefings/pages/06ghginitialstrategy.aspx, acceded on 31/01/20.

[4] T. Bruckner, L. Fulton, E. Hertwich, A. McKinnon et al., Technology-specific Cost and Performance Parameters. Available on line : https://www.ipcc.ch/site/assets/uploads/2018/02/ipcc_wg3_ar5_annex-iii.pdf , page 14. Accessed on 31/01/20.

[5] : Ademe, Information CO2 des prestations de transport - Application de l’article L. 1431-3 du code des transports - Guide méthodologique, Available online :

https://www.ademe.fr/sites/default/files/assets/documents/86275_7715-guide-information-co2-transporteurs.pdf , page 65. Accessed on 31/01/20.

Otherwise the comparison is not fully fair. A port hopping schedule might also consider different speeds as well as risks, therefore the total environmental burden of the fleet, should also consider different speeds among arcs of the network as well as seasonal effects, i.e. regions that face higher risks of icing and therefore or restricted navigation, etc

Absolutely, thanks for your precious comment. However, at this pre-feasibility stage of the study, general ratios can be taken into account. It would obviously be necessary to take into account the particular conditions of navigability according to the depth of the sea, the evolution over the year of the share and the ice consistency to refine this pre-feasibility study into a study of potential. We have added, in blue: « This first step in determining the relevance of developing an electric vessel fleet operating on the NSR should continue with more precise studies, including in particular the different speeds along the route as well as seasonal effects. " before conclusion part.

Round 2

Reviewer 2 Report

well done 

Reviewer 3 Report

Points taken!